# The influences of El Niño–Southern Oscillation on tropospheric ozone in CMIP6 models

Thanh Le[1,2★], Seon-Ho Kim[2], Jae-Yeong Heo[2] and Deg-Hyo Bae[2★]

[1]Key Laboratory of Meteorological Disaster, Ministry of Education (KLME)/Collaborative Innovation Center on Forecast and Evaluation of Meteorological Disasters (CIC-FEMD), Nanjing University of Information Science and Technology, Nanjing 210044, China
[2]Department of Civil and Environmental Engineering, Sejong University, Seoul 05006, Republic of Korea

★Corresponding author(s): Thanh Le (levinhthanh.lvt@gmail.com) and Deg-Hyo Bae (dhbae@sejong.ac.kr)

**Abstract.** Ozone in the troposphere is a greenhouse gas and a pollutant, hence, additional understanding of the drivers of tropospheric ozone evolution is essential. The El Niño–Southern Oscillation (ENSO) is a main climate mode and may contribute to the variations of tropospheric ozone. Nevertheless, there is uncertainty regarding the causal influences of ENSO on tropospheric ozone under warming environment. Here, we investigated the links between ENSO and tropospheric ozone using Coupled Modeling Intercomparison Project Phase 6 (CMIP6) data over the period 1850-2014. Our results show that ENSO impacts on tropospheric ozone are primarily found over oceans, while the signature of ENSO over continents is largely nonsignificant. Springtime surface ozone is more sensitive to ENSO compared to other seasons. The response of ozone to ENSO may vary depending on specific air pressure levels in the troposphere. These responses are weak in the middle troposphere and are stronger in the upper and lower troposphere. There is high consistency across CMIP6 models in simulating the signature of ENSO on ozone over the lower, middle and upper troposphere. While the response of tropical tropospheric ozone to ENSO is in agreement with previous works, our results suggest that ENSO impacts on tropospheric ozone over the northern North Pacific, American continent, and over the mid-latitude regions of the southern Pacific, Atlantic, and Indian oceans might be more significant than previously understood.

**Keywords:** El Niño–Southern Oscillation; historical simulations; tropospheric ozone; interactive chemistry; CMIP6.

## 1 Introduction

Ozone in the troposphere is an important greenhouse gas and pollutant (Archibald et al., 2020; Cooper et al., 2010; Wang et al., 2022). Tropospheric ozone has detrimental effects on human health and ecosystems (Fleming et al., 2018; Franz et al., 2018; Gaudel et al., 2020; Lu et al., 2019; Oliver et al., 2018; Peron et al., 2021; Roberts et al., 2022; Schauberger et al., 2019). Changes in atmospheric ozone may also affect radiative forcing and have effects on climate (Gauss et al., 2006; Myhre et al., 2013).

The El Niño–Southern Oscillation (ENSO) is the major mode of climate variability with global impacts (Bjerknes, 1969; Cai et al., 2020; McPhaden et al., 2006) and is expected to affect variations of global tropospheric ozone. ENSO-induced

changes in climate and meteorological conditions (Le and Bae, 2022a; Lu et al., 2019; Yeh et al., 2018) lead to impacts on ecosystems and the production/removal of ozone in soil, plant, and the water cycle, and the transport of ozone (Ganzeveld et al., 2009; Lin et al., 2019; Neu et al., 2014). In particular, ENSO drives changes in tropospheric and stratospheric circulations which can alter the tropospheric ozone variations (Daskalakis et al., 2022; Domeisen et al., 2019; Koumoutsaris et al., 2008; Lin et al., 2015; Neu et al., 2014; Olsen et al., 2019; Oman et al., 2013; Zeng and Pyle, 2005; Ziemke et al., 2015). In addition, ENSO was revealed to exhibit influences on tropospheric ozone concentrations in many regions by modulating local meteorological conditions (Doherty et al., 2006; Jeong et al., 2023; Jiang and Li, 2022; Oman et al., 2011; Peiro et al., 2018; Rowlinson et al., 2019; Wie et al., 2021; Xu et al., 2017; Yang et al., 2022; Zhang et al., 2015).

Nevertheless, there are uncertainties regarding the causal effects of ENSO on global tropospheric ozone. For instance, while the response of tropospheric ozone to ENSO over the mid-latitude regions remains elusive (Lu et al., 2019; Olsen et al., 2016), further understanding of ENSO impacts on ozone concentrations at multiple air pressure levels in the troposphere is necessary. Moreover, a causal analysis (Le et al., 2022; Le and Bae, 2022b) that takes into account the confounding impacts of other climate modes on the relationship between ENSO and tropospheric ozone is lacking. While the response of tropospheric ozone to ENSO can be interpreted by changes in ENSO-related atmospheric circulation (Lu et al., 2019; Sekiya and Sudo, 2012; Ziemke and Chandra, 2003), these changes might be influenced by other climate modes (Cai et al., 2019; Le et al., 2020b). Despite the high spatial and temporal variability of tropospheric ozone, there are limited observations of past ozone changes at the global scale (Dragani, 2011; Ebojie et al., 2016; Gaudel et al., 2018; Young et al., 2018). Hence, Earth system models remain valuable tools to understand the evolution of tropospheric ozone and the interactions between tropospheric ozone and regional climate (Archibald et al., 2020; Collins et al., 2017; Young et al., 2018). Datasets from Coupled Modeling Intercomparison Project Phase 6 (CMIP6) models provide an important source to better identify the effects of ENSO on global tropospheric ozone.

In the present study, we evaluated the causal impacts of ENSO on tropospheric ozone at the global scale using data from the historical simulations of CMIP6 models. We also discussed the coherency across CMIP6 models in reproducing the connection between ENSO and tropospheric ozone.

## 2    Materials and Methods

### 2.1    Datasets

We used monthly data of mole fraction of ozone in air (i.e., variable 'o3') at different air pressure levels (i.e., 1000, 850, 500, and 300 hPa). The CMIP6 models with ozone data available for the historical simulations (Eyring et al., 2016) over the 1850-2014 period are listed in Table 1. In Table 1, the models equipped with an Atmospheric Chemistry module are fully coupled where the chemistry scheme is associated with the physics of the atmospheric model, allowing for comprehensive consideration of interactions between climate variations, interactive chemistry, and carbon cycle (Emmons et al., 2020;

Michou et al., 2020; Wu et al., 2019). The use of various model outputs reduces the uncertainty of the connections between ENSO and tropospheric ozone.

There are biases in simulating tropospheric ozone variations in the models (Griffiths et al., 2021; Turnock et al., 2020; Young et al., 2018). For instance, CMIP6 models may underestimate ozone levels in the Southern Hemisphere and overestimate ozone levels in the Northern Hemisphere compared to observational data of recent past (Griffiths et al., 2021; Turnock et al., 2020; Young et al., 2018). However, CMIP model outputs are still helpful to investigate the effects of ENSO on tropospheric ozone (Archibald et al., 2020; Young et al., 2018). For example, the simulation of tropospheric ozone in CESM2 models is improved in comparison to previous model versions (Emmons et al., 2020). In addition, CMIP6 models are capable of simulating long-term changes in surface ozone levels and recent increasing trends in tropospheric ozone (Griffiths et al., 2021; Turnock et al., 2020).

We employed monthly sea level pressure (SLP) and sea surface temperature (SST) to calculate the time series of the major climate modes (see also section Methods 2.2).

**Table 1.** List of CMIP6 models used in this study.

| No. | Model name | Modelling center, country | Atmospheric Chemistry model |
|-----|-----------|---------------------------|----------------------------|
| 1 | BCC_CSM2_MR | BCC, China | None |
| 2 | BCC_ESM1 | BCC, China | BCC-AGCM3-Chem |
| 3 | CESM2 | NCAR, United States | MOZART-T1 |
| 4 | CESM2_FV2 | NCAR, United States | MOZART-T1 |
| 5 | CESM2_WACCM | NCAR, United States | MOZART-T1 |
| 6 | CESM2_WACCM_FV2 | NCAR, United States | MOZART-T1 |
| 7 | CNRM_CM6_1 | CNRM-CERFACS, France | OZL_v2 |
| 8 | CNRM_CM6_1_HR | CNRM-CERFACS, France | OZL_v2 |
| 9 | CNRM_ESM2_1 | CNRM-CERFACS, France | REPROBUS-C_v2 |
| 10 | IPSL_CM6A_LR | IPSL, France | None |
| 11 | MPI_ESM_1_2_HAM | MPI-M, Germany | Sulfur chemistry (unnamed) |
| 12 | MPI_ESM1_2_LR | MPI-M, Germany | None |

## 2.2 Methods

We assess the possibility of the impacts of ENSO on tropospheric ozone based on the approach employed in recent studies (Le and Bae, 2020, 2022a). This method was established using a multivariate predictive model to assess the probability for

the absence of Granger causal effects of ENSO on ozone concentrations. In the computations, we considered the confounding impacts of other major climate modes (i.e., the dipole mode index (DMI; Saji et al., 1999), the Southern Annular Mode (SAM; e.g., Cai et al., 2011), and the North Atlantic Oscillation (NAO; e.g., Hurrell et al., 2003)). Given that the climate changes in the Indian and Atlantic oceans can affect the tropical Pacific (Cai et al., 2019; Ha et al., 2017a; Le et al., 2020a; Le and Bae, 2019), and modify the connections between ENSO and ozone concentrations, these analyses provide a realistic estimate for the response of ozone concentrations to ENSO.

We use the following multivariate predictive model (e.g., Stern and Kaufmann 2013, Mosedale *et al* 2006) to estimate the causal links between the ENSO and ozone concentration:

$$X_t = \sum_{i=1}^{p} \alpha_i X_{t-i} + \sum_{i=1}^{p} \beta_i Y_{t-i} + \sum_{j=1}^{m} \sum_{i=1}^{p} \delta_{j,i} Z_{j,t-i} + \varepsilon_t \tag{1}$$

where $X_t$ is the annual mean (or seasonal mean) ozone concentration for year $t$, $Y_t$ is the ENSO index, and $Z_{j,t}$ is the confounding factor $j$ for year $t$. In the predictive model shown in equation 1, while estimating the influence of $Y$ on $X$ (i.e., the contribution of the term $\sum_{i=1}^{p} \beta_i Y_{t-i}$ in predicting $X$), the contribution of past $X$ events are already taken into account by adding the term $\sum_{i=1}^{p} \alpha_i X_{t-i}$. Thus, the causal influence of $Y$ on $X$, if detected, is robust and the contribution of past $X$ events are already considered in our analyses. Here, $m$ is number of confounding factors and $p \geq 1$ is the order of the multivariate predictive model. The optimal order $p$ is computed by minimizing the Schwarz criterion or the Bayesian information criterion (Schwarz, 1978). The optimal orders might be different for each model.

The ENSO index was computed as the average sea surface temperature (SST) anomalies in the Niño 3.4 area (120–170°W; 5°N–5°S) in boreal winter (December–January–February, DJF). Confounding factors (i.e., the dipole mode index (DMI; Saji et al., 1999), the Southern Annular Mode (SAM) and the North Atlantic Oscillation (NAO; e.g., Hurrell et al., 2003)) may have effects on the connections between ENSO and ozone concentration. The DMI was given as the difference in boreal fall (September–October–November, SON) SST anomalies between two Indian Ocean regions of the western pole (50–70°E; 10°N–10°S) and southeastern pole (90–110°E; 0°N–10°S). The SAM (Cai et al., 2011) was calculated as the first empirical orthogonal function (EOF) of the boreal summer (June–July–August, JJA) sea level pressure (SLP) anomalies for the region of 40–70°S. The NAO index is computed as the EOF of boreal winter (DJF) SLP anomalies in the North Atlantic area (90°W-40ºE, 20º-70ºN). In this study, the confounding factors are limited to three major climate modes (i.e., DMI, SAM and NAO) as these modes are crucial to global climate variability on interannual time scales (Delworth et al., 2016; Hurrell et al., 2003; Kripalani et al., 2009; Luo et al., 2012; Raphael and Holland, 2006). Furthermore, alterations in these climate modes may influence the variations of ENSO (Cai et al., 2019; Ha et al., 2017b; Le et al., 2020b; Le and Bae, 2019).

We estimate the probability of no Granger causality by applying a test of Granger causality (Le and Bae, 2020; Mosedale et al., 2006; Stern and Kaufmann, 2013) for the multivariate predictive model shown in equation 1. For computing the degree of uncertainty, we followed recent guidance (Stocker et al., 2013) and utilized the terms 'very unlikely', 'unlikely', 'likely' for the 0–10%, 0–33%, and 66–100% probability of the likelihood of the outcome, respectively. For example, if the *p*-value

is less than 0.33, the result indicates that ENSO is unlikely to display no Granger causality on ozone concentration. In this instance, we conclude that ENSO has 'causal effect' on ozone concentration.

## 3    Results

Figure 1 depicts the models' mean map of ozone concentrations at different air pressure levels for the period 1850-2014 of the CMIP6 historical simulations. As we will show in Figure 3, the models without the Atmospheric Chemistry module (i.e., 4 models numbered 1, 10, 11 and 12 in Table 1) exhibit distinct outcomes of ENSO impacts compared to both the models' mean and the remaining models. Hence, it's important to note that the models' mean is solely derived from the results obtained from all the models equipped with an Atmospheric Chemistry module (i.e., 8 models numbered from 2 to 9 in Table 1). In the middle and lower troposphere, ozone is higher in the northern hemisphere compared to the southern hemisphere (Figure 1). The agreement between models fluctuates at different air pressure levels. For example, in the upper troposphere (i.e., at 300 hPa pressure level), high consistency across the model is found in the mid-latitude regions, while this consistency is lower in the tropics and polar regions (Figure 1a). In the middle troposphere (i.e., at 500 hPa pressure level), the models' agreement is mainly found in the northern hemisphere (Figure 1b). The simulations of near-surface ozone (i.e., at 850 hPa) are consistent over parts of the northern hemisphere (Figure 1c), while the models' agreement is low in reproducing surface ozone (i.e., at 1000 hPa) for most regions (Figure 1d). The standard deviation is normally higher in the tropics and much of the southern hemisphere compared to other regions (Figure S1).

Figure 2 displays the causal effects of ENSO on global ozone concentrations for the historical period 1850-2014. In Figure 2, we show that the response of ozone to ENSO may vary depending on specific air pressure level. For instance, ENSO impacts on ozone in the upper troposphere (i.e., at 300 hPa pressure level) can be observed over the tropics, parts of the Pacific Ocean, South America and North America (Figure 2a). In these areas, ENSO is unlikely to exhibit no causal influences on ozone concentrations (i.e., $p$-values were lower than 0.33). Further analysis (not shown) indicates that the patterns of ENSO impacts on ozone at 250 hPa are similar to those at 300 hPa. This implies that the response of ozone variation to ENSO might remain consistent across the upper troposphere, the tropopause, and the lower stratosphere. The response of ozone to ENSO in the middle troposphere (i.e., at 500 hPa pressure level) is found over the tropical Pacific and Atlantic Oceans (Figure 2b). We observe more significant impacts of ENSO on ozone in the lower troposphere compared to the upper and middle troposphere. Specifically, tropical ozone concentrations at near surface (i.e., at 850 hPa) and surface (i.e., at 1000 hPa) levels appear to be sensitive to ENSO (Figures 2c and d). In these regions, ENSO is very unlikely to exhibit no causal influences on ozone concentrations (i.e., $p$-values were lower than 0.1). In addition, ENSO impacts on surface ozone can be found over part of the northern North Pacific and mid-latitude regions in the southern hemisphere (Figure 2d). While the signature of ENSO on ozone variations is generally weak over continents of the lower and middle troposphere (Figures 2b-d), this signature is, however, stronger in the upper troposphere over North America (Figure 2a). These results imply that the effects of ENSO on tropospheric ozone over lands are nonsignificant for most regions.

Differences between CMIP6 models in replicating the influences of ENSO on surface (1000 hPa) ozone are shown in Figure 3. Similar results for other pressure levels (300, 500, and 850 hPa) are presented in Figures S2-S4. As revealed in Figure 3, several models (i.e., BCC_CSM2_MR, IPSL_CM6A_LR, and MPI_ESM1_2_LR) may not reproduce the significant influences of ENSO on surface ozone over the tropical Pacific and Indian Oceans as described in Figure 2d. The models IPSL_CM6A_LR and MPI_ESM1_2_LR may underestimate the response of surface ozone to ENSO over the mid-latitude regions in the southern hemisphere compared to other models. The agreement between models for the impacts of ENSO on surface ozone is low over continents (Figure 2d), partly due to the discrepancy in simulating ozone variability (Figure 1d). While there are biases in simulating the response of surface ozone to ENSO (Figure 2d), these responses in the middle and upper troposphere appear to be more consistent across models (Figures 2a-c, S2-4).

Springtime surface ozone is more sensitive to ENSO compared to other seasons (Figure 4). In particular, the clear response of springtime surface ozone over the tropics, the high-latitude north Pacific and the mid to high-latitude of the southern hemisphere can be observed (Figure 4a). The impacts of ENSO on surface ozone of other seasons are limited (e.g., over the tropical Pacific and part of southern North America, Figure 4b-d). The results for other air pressure levels (300, 500, and 850 hPa) are shown in Figures S5-S7. We note that the response of springtime ozone at higher pressure levels is weaker compared to springtime surface ozone for most regions, except for the upper troposphere over east Asia, northern South America and northwestern North America (Figure S5a). Consistent with the results illustrated in Figure 2b, the impacts of ENSO on seasonal ozone in the middle troposphere (500 hPa) are mainly significant over the tropics and part of northern North Pacific (Figure S6).

## 4    Discussion and conclusions

The effects of ENSO on tropospheric ozone over the tropical Pacific (Figures 2-4) show agreement with previous works (Chandra et al., 2007; Peiro et al., 2018). ENSO causes changes in the tropospheric ozone budget over the tropical Pacific by modulating the Walker circulation (Chandra et al., 2007), wind systems (Cai et al., 2021; Le and Bae, 2020; Yeh et al., 2018), and inducing biomass burning (Chandra et al., 2009; Le et al., 2022). Further, significant ENSO impacts on tropical ocean regions described in Figures 2-4 are in agreement with recent works (Olsen et al., 2016; Wespes et al., 2017) using satellite data.

Despite the limited consensus among models in replicating ozone levels in the lower troposphere, and a high standard deviation particularly in tropical regions, (Figures 1 and S1), we observed noteworthy effects of ENSO on lower tropospheric ozone (Figure 2). These results exhibit a degree of independence and are not contradictory. This is because the models' mean of annual ozone is calculated over the entire 1850-2014 period, whereas the assessment of the relationship between the ENSO and annual ozone is conducted on a year-to-year basis. Furthermore, variations in ozone are also influenced by factors beyond ENSO, including other major climate modes, cyclones, and local emissions of ozone precursors such as nitrogen oxides ($NO_x$), volatile organic compounds, and carbon monoxide (CO). Biases in simulating

these factors contribute to the inconsistencies of ozone in the models, although there is consensus in simulating the connection between ENSO and ozone.

The significant impacts of ENSO on ozone in the upper troposphere (300 hPa) over the southern and western North America (Figures 2a and S5a) might be associated with the transport of ozone from east Asia (Cooper et al., 2010; Doherty, 2015; Lin et al., 2015). These impacts can be explained by the modulation of ENSO on springtime upper tropospheric ozone over east

Asia (Figure S5a) and the connection between ENSO and the North Pacific Oscillation (Kug et al., 2020). However, these impacts cannot reach the surface levels (Figures 2c-d, and S7a), consistent with recent work (Lin et al., 2015).

We note that the models without the Atmospheric Chemistry module (BCC_CSM2_MR, IPSL_CM6A_LR, MPI_ESM_1_2_HAM, and MPI_ESM1_2_LR; See Table 1) provide different results of ENSO impacts compared to the rest models (Figures 3). In these models, ozone variations are prescribed using observational data (Lurton et al., 2020; Wu et al.,

2019), and it is expected that the response of ozone variation to atmospheric circulation and ENSO is not significant. Hence, improvement of the Atmospheric Chemistry module in the models may provide further understanding of the connection between ENSO and ozone variations.

The robust response of lower tropospheric ozone to ENSO is associated with ENSO-induced changes in the atmospheric circulation (Oman et al., 2011) and this response is particularly prominent over the tropics (Figures 2c and d). However, this

response appears to be weaker over the middle and upper troposphere (Figures 2a and b). The weak impacts of ENSO on the mid-level tropospheric ozone (i.e., 500 hPa level, described in Figures 2b) might be due to the strong exchange between stratospheric ozone and middle to upper tropospheric ozone (Liu et al., 2017; Meul et al., 2018; Neu et al., 2014; Williams et al., 2019). The more pronounced reaction of upper tropospheric ozone to ENSO in comparison to middle tropospheric ozone could be attributed to the influence of ENSO on deep convective transport and the interconnected relationship between

ENSO and the North Pacific Oscillation (Cai et al., 2019; Gaudel et al., 2020; Kug et al., 2020).

Several models showed ENSO effects on tropospheric ozone over Antarctica with a low agreement between models (Figures 2-4). These impacts might be associated with the signature of ENSO on stratospheric ozone anomalies over Antarctica (Li et al., 2021; Lin and Qian, 2019).

Given that the tropospheric ozone burden and the ozone-induced impacts may increase in some regions in the future

(Doherty et al., 2013; Franz and Zaehle, 2021; Gaudel et al., 2020; Griffiths et al., 2021; Verstraeten et al., 2015; Zanis et al., 2022), further analyses of ENSO impacts on tropospheric ozone in future climate projections are necessary. In addition, as the tropopause may vary depending on different latitudes (Griffiths et al., 2021), it is essential to conduct further analyses that specifically address the impacts of ENSO on ozone concentrations across the upper troposphere, the tropopause, and the lower stratosphere.

## Acknowledgments

The authors thank editor Graciela Raga and the anonymous reviewers for their valuable comments and suggestions. We acknowledge the World Climate Research Programme, which through its Working Group on Coupled Modelling, coordinated and promoted CMIP6. We thank the climate modelling groups (listed in Table 1) for producing and making available their model output, the Earth System Grid Federation (ESGF) for archiving the data and providing access, and the multiple funding agencies who support CMIP6 and ESGF. Thanh Le is supported by the Startup Foundation for Introducing Talent of Nanjing University of Information Science and Technology (NUIST) and by the National Research Foundation of Korea (NRF) grant funded by the Korea government (MSIT) (Grant No. 2021R1G1A1004389).

## Data Availability

The data that support the findings of this study are openly available at the following website: https://esgf-node.llnl.gov/search/cmip6/.

## Author contribution

TL designed the study, performed the data analysis, and wrote the manuscript. SHK, JYH and DHB contributed to the interpretation of results and the writing of the manuscript.

## Competing interests

The authors declare that they have no conflict of interest.

## Financial support

This work is by the Startup Foundation for Introducing Talent of Nanjing University of Information Science and Technology (NUIST) and supported by the National Research Foundation of Korea (NRF) grant funded by the Korea government (MSIT) (Grant No. 2021R1G1A1004389).

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

485

490

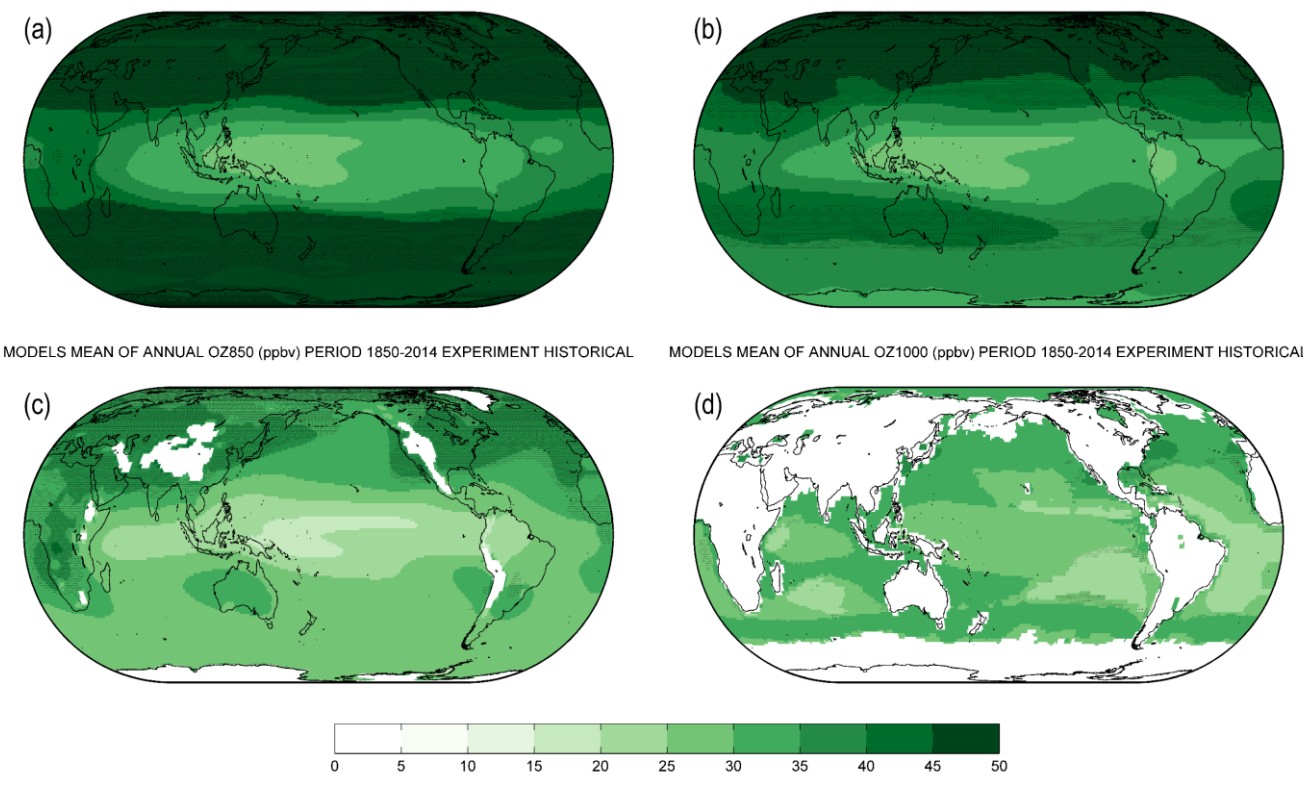

**Figure 1.** Multi-model mean map of annual mean ozone concentrations (ppbv) for the historical experiment over the 1850-2014 period at 300 hPa (a), 500 hPa (b), 850 hPa (c) and 1000 hPa (d) pressure levels, respectively. Stippling indicates that at least 70% of total models show agreement on the mean ozone concentrations of all models at given grid point. The agreement of an individual model is identified when the difference between the selected model's ozone concentrations and the multi-model mean ozone concentrations is less than one standard deviation of the multi-model mean ozone concentrations.

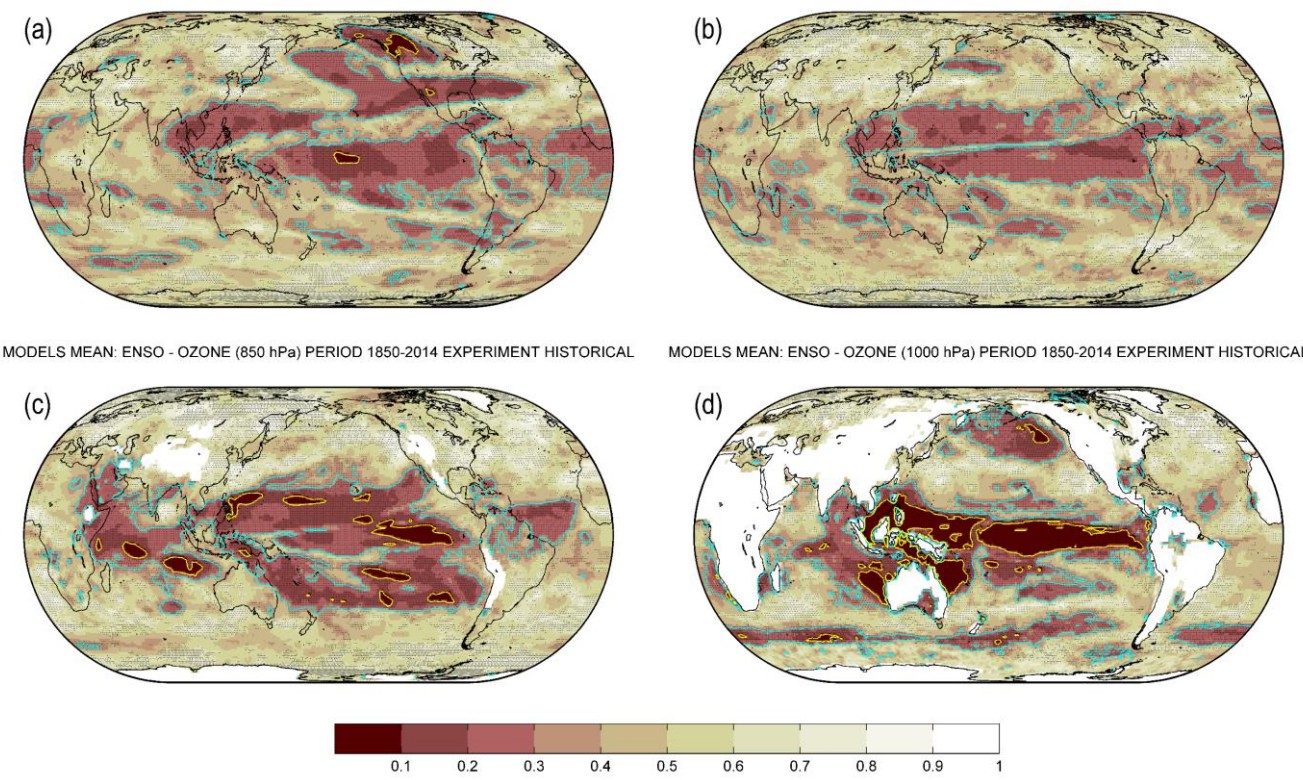

**Figure 2.** Map of multi-model mean probability for the absence of Granger causality from ENSO to annual ozone concentrations for the historical experiment over the 1850-2014 period at 300 hPa (a), 500 hPa (b), 850 hPa (c) and 1000 hPa (d) pressure levels, respectively. Stippling indicates that at least 70% of total models show agreement on the mean probability of all models at given grid point. The agreement of an individual model is identified when the difference between the selected model's probability and the multi-model mean probability is less than one standard deviation of the multi-model mean probability. The cyan and yellow contour lines denote *p*-value = 0.33 and 0.1, respectively. Brown shades indicate low probability of no Granger causality. ENSO = El Niño–Southern Oscillation.

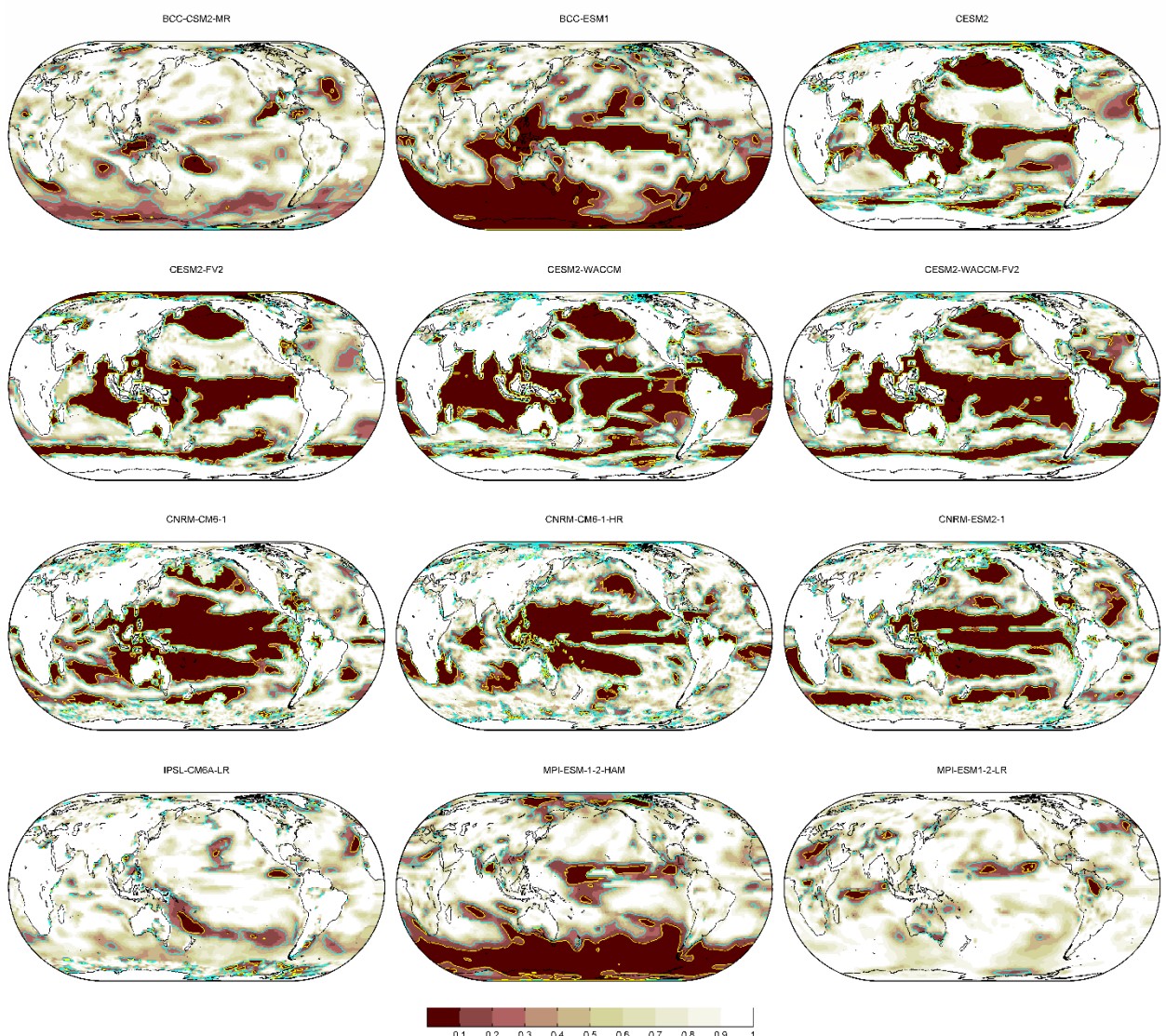

**Figure 3.** Probability of no Granger causality from ENSO to annual ozone concentrations at 1000 hPa pressure level for the historical experiment over the 1850-2014 period of 12 individual models (see Table 1). The yellow and cyan contour lines denote $p$-value = 0.1 and 0.33, respectively. Brown shades imply a low probability of no Granger causality. ENSO: El Niño–Southern Oscillation.

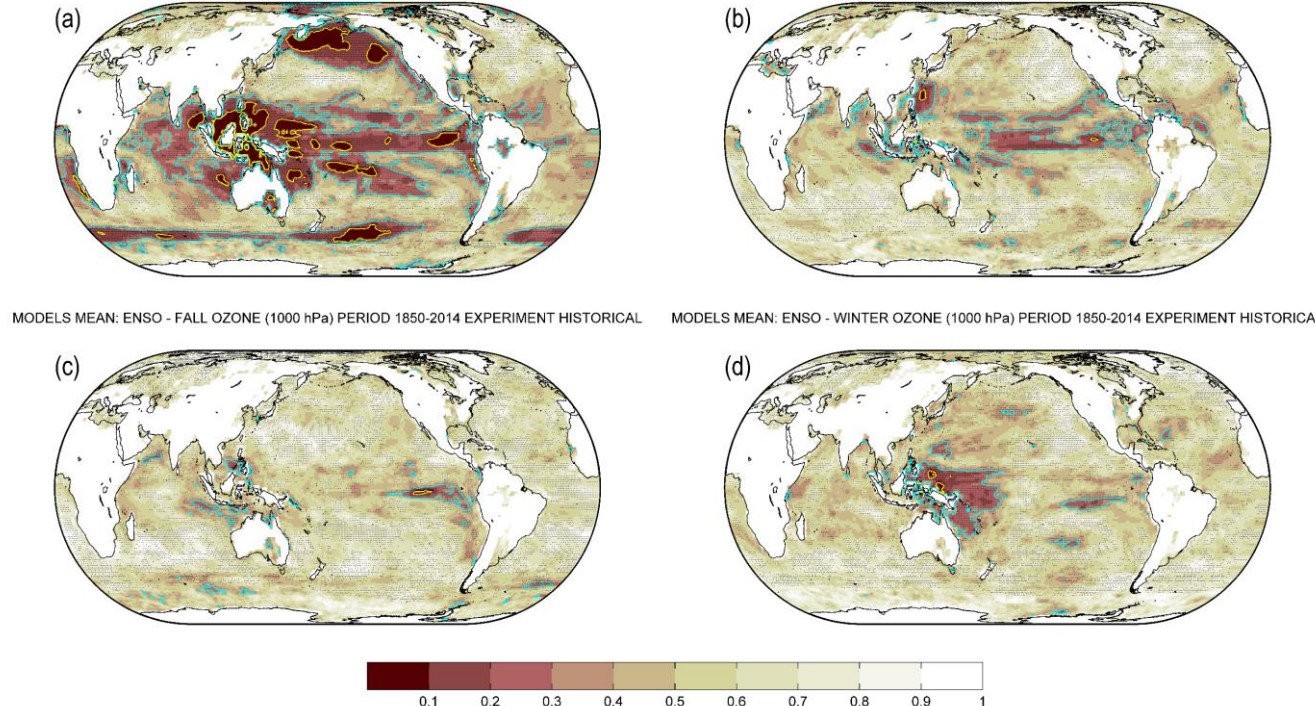

**Figure 4.** Multi-model mean probability map of no Granger causality from ENSO in boreal winter [D(*t*)JF(*t+1*); *t* indicates year *t*] to seasonal mean ozone concentrations at 1000 hPa pressure level over the period 1850-2014. (a) Spring [MAM(*t+1*)]. (b) Summer [JJA(*t+1*)]. (c) Fall [SON(*t+1*)]. (d) Winter [D(*t+1*)JF(*t+2*)]. Stippling signifies that at least 70% of total models show agreement on the mean probability of all models at a given grid point. The cyan contour line signifies *p*-value = 0.33. Brown shades imply a low probability of no Granger causality. ENSO: El Niño–Southern Oscillation. MAM: March- April-May. JJA: June-July-August. SON: September-October-November. DJF: December-January-February.