# Peer review of "The influences of El Niño–Southern Oscillation on tropospheric ozone in CMIP6 models"

_EGUsphere, 2023_

## Referee Comment (RC1)

**The Influences of El-Nino Southern Oscillation on tropospheric ozone in CMIP6 models**

The manuscript titled **"The Influences of El-Nino Southern Oscillation on tropospheric ozone in CMIP6 models"** by Thanh Le et al., attempted to mainly investigate the influence of El Niño Southern Oscillation (ENSO) on the global tropospheric ozone variations by making use of the historical simulations of the earth system climate models of CMIP6 by multivariate predictive model approach. The authors find that ENSO response to ozone is stronger in the upper and lower troposphere and these findings emphasized that the impact of the ENSO on the tropospheric ozone over mid latitude regions like southern Pacific, Atlantic and Indian oceans are significant then previously understood. Overall, this paper is well written and their findings are potentially helpful in understanding the effect of changes in the tropospheric ozone and future climate projections. I would like to recommend an acceptation after the following comments are addressed.

**Major comments**

1. The Pacific decadal oscillation which is also one of the main climate mode that can affect ENSO and indeed on the ozone concentrations. The authors didn't explain why other climate modes are not considered and   why only three (Dipole mode Index, Southern Annual Mode and North Atlantic Oscillation) climate modes.

2. Try to elaborate mainly the common schemes in the Atmospheric Chemistry Modules that are in the models (other than the three models BCC_CSM2_MR, IPSL_CM6A_LR and MPI_ESM1_2_LR) as the behavior of these models in connection to the response of ENSO on ozone variation is similar.

3. The Text S1 which explains about the method that has been adopted should be mentioned under the method section 2.2 rather than in the supplement. It helps the reader to have a quick through of the methodology adopted in the study.

4. Why did you consider only 1000 hPa, 850 hPa, 500 hPa and 300 hPa ? Are these pressure levels enough to represent the respective atmospheric region of the atmosphere (like middle troposphere, upper troposphere). As ENSO is responsible for changes in winds and circulation patterns. It is also expected to have impact on the transport of ozone from the lower troposphere to upper troposphere and lower stratosphere. It would be interesting if you can check if the features are same in the upper levels (above 300 hPa just below the tropopause)

**Specific Comments**

**Line Nos.:42:43**: Did you check if the findings obtained using CMIP6 and CMIP5 ?  If so where did you find the changes that resulted in the current result?

**Line Nos.: 51:** The list of the models mentioned in Table S1 should be shifted to the main manuscript instead of supplement.

**Line Nos. 53:55**: The authors are suggested to explain little more on the findings of the cited papers rather than just citing the paper.

The Figures can be of more clarity (mainly the stippling in figures are not at all visible (for example Figure 1 (a)) are not visible clearly, The titles in the Figure 3 should be made little big)

---

## Author Comment (AC1)

**1 Response to Reviewer #1's comments**

**1.1**  1. The Pacific decadal oscillation which is also one of the main climate mode that can affect ENSO and indeed on the ozone concentrations. The authors didn't explain why other climate modes are not considered and why only three (Dipole mode Index, Southern Annual Mode and North Atlantic Oscillation) climate modes.

**Response:** We thank the reviewer for raising this point. We agree that the Pacific decadal oscillation (PDO) is an important climate mode. However, as we mainly focus on the impacts of ENSO on interannual time scale, we have not included the PDO in the analysis.

We added the following sentences to Section 2.2 to clarify this point:

"In this study, the confounding factors are limited to three major climate modes (i.e., DMI, SAM and NAO) as these modes are crucial to global climate variability on interannual time scales (Delworth et al., 2016; Hurrell et al., 2003; Kripalani et al., 2009; Luo et al., 2012; Raphael and Holland, 2006). Furthermore, alterations in these climate modes may influence the variations of ENSO (Cai et al., 2019; Ha et al., 2017; Le et al., 2020; Le and Bae, 2019)."

**1.2**  2. Try to elaborate mainly the common schemes in the Atmospheric Chemistry Modules that are in the models (other than the three models BCC_CSM2_MR, IPSL_CM6A_LR and MPI_ESM1_2_LR) as the behavior of these models in connection to the response of ENSO on ozone variation is similar.

**Response:** We thank the reviewer for raising this point. We added the following sentences to Section 2.1 and Section 4 to clarify this point:

"In Table 1, the models equipped with an Atmospheric Chemistry module are fully coupled where the chemistry scheme is associated with the physics of the atmospheric model, allowing for comprehensive consideration of interactions between climate variations, interactive chemistry, and carbon cycle (Emmons et al., 2020; Michou et al., 2020; Wu et al., 2019)."

"In these models, ozone variations are prescribed using observational data (Lurton et al., 2020; Wu et al., 2019), and it is expected that the response of ozone variation to atmospheric circulation and ENSO is not significant."

1.3   3. The Text S1 which explains about the method that has been adopted should be mentioned
under the method section 2.2 rather than in the supplement. It helps the reader to have a
quick through of the methodology adopted in the study.
**Response:** We thank the reviewer for this suggestion. We moved Text S1 to Section 2.2 of the
main text.

1.4   4. Why did you consider only 1000 hPa, 850 hPa, 500 hPa and 300 hPa ? Are these pressure
levels enough to represent the respective atmospheric region of the atmosphere (like middle
troposphere, upper troposphere). As ENSO is responsible for changes in winds and
circulation patterns. It is also expected to have impact on the transport of ozone from the
lower troposphere to upper troposphere and lower stratosphere. It would be interesting if you
can check if the features are same in the upper levels (above 300 hPa just below the
tropopause)
**Response:** We thank the reviewer for raising this point. In our opinion, the selected pressure levels
can represent much of the atmosphere as supported by the results described in Figure 2. In Figure
2, there might be distinct impacts of ENSO on ozone over the lower, middle, and upper
troposphere.
Below we show the analysis at 250 hPa. At this pressure level, the regions from 60N-90N are in
the lower stratosphere, while the regions from 90S-60N are in the upper troposphere (Griffiths et
al., 2021). Figure R1 below shows that the pattern of ENSO impacts for the analysis at 250 hPa is
similar to the analysis at 300 hPa. Hence, we conclude that there is no significant change in ENSO
impacts on ozone at the tropopause, though additional analyses might give clearer answer.
We added the following sentences to Sections 3 and 4 to discuss this point:
"Further analysis (not shown) indicates that the patterns of ENSO impacts on ozone at 250 hPa are
similar to those at 300 hPa. This implies that the response of ozone variation to ENSO might
remain consistent across the upper troposphere, the tropopause, and the lower stratosphere."
"In addition, as the tropopause may vary depending on different latitudes (Griffiths et al., 2021),
it is essential to conduct further analyses that specifically address the impacts of ENSO on ozone
concentrations across the upper troposphere, the tropopause, and the lower stratosphere."

MODELS MEAN: ENSO - OZONE (250 hPa) PERIOD 1850-2014 EXPERIMENT HISTORICAL

[Figure]

MODELS MEAN: ENSO - OZONE (300 hPa) PERIOD 1850-2014 EXPERIMENT HISTORICAL

[Figure]

**Figure R1.** Map of multi-model mean probability for the absence of Granger causality from ENSO to annual ozone concentrations for the historical experiment over the 1850-2014 period at 250 hPa (upper) and 300 hPa (lower).

1.5    Line Nos.:42:43: Did you check if the findings obtained using CMIP6 and CMIP5 ? If so
where did you find the changes that resulted in the current result?
**Response:** We thank the reviewer for raising this point. We have not tried to add the analyses of
CMIP5 models because there is limitations in these models (Emmons et al., 2020; Michou et al.,
2020).
Further explanation is added to Section 2.1:
"For example, the simulation of tropospheric ozone in CESM2 models is improved in comparison
to previous model versions (Emmons et al., 2020). In addition, CMIP6 models are capable of
simulating long-term changes in surface ozone levels and recent increasing trends in tropospheric
ozone (Griffiths et al., 2021; Turnock et al., 2020)."

1.6    Line Nos.: 51: The list of the models mentioned in Table S1 should be shifted to the main
manuscript instead of supplement.
**Response:** We thank the reviewer for this suggestion. We moved Table S1 to Section 2.1 of the
main text.

1.7    Line Nos. 53:55: The authors are suggested to explain little more on the findings of the cited
papers rather than just citing the paper.
**Response:** We thank the reviewer for raising this point. We added the following sentences to
Section 2.1 to clarify this point:
"For instance, CMIP6 models may underestimate ozone levels in the Southern Hemisphere and
overestimate ozone levels in the Northern Hemisphere compared to observational data of recent
past (Griffiths et al., 2021; Turnock et al., 2020; Young et al., 2018)."
"For example, the simulation of tropospheric ozone in CESM2 models is improved in comparison
to previous model versions (Emmons et al., 2020). In addition, CMIP6 models are capable of
simulating long-term changes in surface ozone levels and recent increasing trends in tropospheric
ozone (Griffiths et al., 2021; Turnock et al., 2020)."

1.8    The Figures can be of more clarity (mainly the stippling in figures are not at all visible (for
example Figure 1 (a)) are not visible clearly, The titles in the Figure 3 should be made little
big)
**Response:** We thank the reviewer for this suggestion. We will provide higher resolution figures.

**References**

[revised manuscript text omitted]

Elem. Sci. Anthr., 6, doi:10.1525/elementa.265, 2018.

---

## Author Comment (AC2)

**2  Response to Reviewer #3's comments**

This study investigated the effect of ENSO on tropospheric ozone over the period 1850-2014, focusing on the 300, 500, 850 and 1000 hPa. The authors also used the probability for the absence of Granger causality from ENSO to ozone concentrations. The topic is interesting. However, before it can be considered for publication, some aspects need more explanation.

2.1  My major concern is that can the current CMIP6 model simulations including the ozone chemistry and it related physical and chemical processes. For example, the first BCC model does not have atmospheric chemistry model (Table S1), how can it predict ozone?

**Response:** We thank the reviewer for raising this point. We agree that several models do not have atmospheric chemistry model. However, it might be useful to include these models in the analysis. The comparison between different models may emphasize the importance of the atmospheric chemistry module. For the models without atmospheric chemistry module, the variations of ozone are prescribed and mainly based on observations (Lurton et al., 2020; Wu et al., 2019).

We added the following sentences to Section 4 to further clarify this point:

"In these models, ozone variations are prescribed using observational data (Lurton et al., 2020; Wu et al., 2019), and it is expected that the response of ozone variation to atmospheric circulation and ENSO is not significant."

2.2  The No.3-6 are all CESM2 model. Do these model configurations predict tropospheric ozone with fully atmospheric chemistry?

**Response:** We thank the reviewer for raising this point.

We added the following sentences to Section 2.1 to further clarify this point:

"In Table 1, the models equipped with an Atmospheric Chemistry module are fully coupled where the chemistry scheme is associated with the physics of the atmospheric model, allowing for comprehensive consideration of interactions between climate variations, interactive chemistry, and carbon cycle (Emmons et al., 2020; Michou et al., 2020; Wu et al., 2019)."

"For example, the simulation of tropospheric ozone in CESM2 models is improved in comparison to previous model versions (Emmons et al., 2020)."

2.3  The MAM4 is the name of aerosol module not the atmospheric chemistry.

**Response:** We thank the reviewer for raising this point. We corrected the model name to
MOZART-T1 (the Model for Ozone and Related chemical Tracers with new tropospheric
chemistry scheme) (Emmons et al., 2020).

2.4   Also, are the simulated ozone in these models evaluated? Some models cannot well
reproduce the global distribution of ozone and some cannot characterize the response of
ozone to ENSO signal shown in observations.

**Response:** The performance of CMIP6 models in simulating ozone was assessed in previous
works (Emmons et al., 2020; Griffiths et al., 2021; Turnock et al., 2020; Young et al., 2018). We
agree with the reviewer that the models still have biases in simulating ozone. However, there is
improvement in the current models.
We described this aspect in the section 2.1 of the original manuscript as below:
"There are biases in simulating tropospheric ozone variations in the models (Griffiths et al., 2021;
Turnock et al., 2020; Young et al., 2018), however, CMIP model outputs are still helpful to
investigate the effects of ENSO on tropospheric ozone (Archibald et al., 2020; Young et al.,
2018)."
We added the following sentences to Section 2.1 to further explain this point:
"For instance, CMIP6 models may underestimate ozone levels in the Southern Hemisphere and
overestimate ozone levels in the Northern Hemisphere compared to observational data of recent
past (Griffiths et al., 2021; Turnock et al., 2020; Young et al., 2018)."
"For example, the simulation of tropospheric ozone in CESM2 models is improved in comparison
to previous model versions (Emmons et al., 2020). In addition, CMIP6 models are capable of
simulating long-term changes in surface ozone levels and recent increasing trends in tropospheric
ozone (Griffiths et al., 2021; Turnock et al., 2020)."

2.5   The conclusions about the effect of ENSO on seasonal ozone in the troposphere can be added
to the abstract.

**Response:** We thank the reviewer for this suggestion. We added the following sentence to the
abstract.
"Springtime surface ozone is more sensitive to ENSO compared to other seasons".

2.6  Line35-40: It is suggested to provide the details of the uncertainties regarding the causal
effects of ENSO on global tropospheric ozone. Although the authors provided some
references, the information from these references should be strengthened.

**Response:** We thank the reviewer for raising this point. We added the following sentences to the
Introduction to further clarify this point:
"Moreover, a causal analysis (Le et al., 2022; Le and Bae, 2022) that takes into account the
confounding impacts of other climate modes on the relationship between ENSO and tropospheric
ozone is lacking. While the response of tropospheric ozone to ENSO can be interpreted by changes
in ENSO-related atmospheric circulation (Lu et al., 2019; Sekiya and Sudo, 2012; Ziemke and
Chandra, 2003), these changes might be influenced by other climate modes (Cai et al., 2019; Le et
al., 2020)."

2.7  The effect of ENSO on ozone in the lower troposphere is more significant than that in the
upper and middle troposphere. Please elaborate the reason.

**Response:** We thank the reviewer for raising this point. We modified the relevant paragraph in
Section 4 to further discuss the different effects of ENSO on ozone at different pressure levels as
below:
"The robust response of lower tropospheric ozone to ENSO is associated with ENSO-induced
changes in the atmospheric circulation (Oman et al., 2011) and this response is particularly
prominent over the tropics (Figures 2c and d). However, this response appears to be weaker over
the middle and upper troposphere (Figures 2a and b). The weak impacts of ENSO on the mid-level
tropospheric ozone (i.e., 500 hPa level, described in Figures 2b) might be due to the strong
exchange between stratospheric ozone and middle to upper tropospheric ozone (Liu et al., 2017;
Meul et al., 2018; Neu et al., 2014; Williams et al., 2019). The more pronounced reaction of upper
tropospheric ozone to ENSO in comparison to middle tropospheric ozone could be attributed to
the influence of ENSO on deep convective transport and the interconnected relationship between
ENSO and the North Pacific Oscillation (Cai et al., 2019; Gaudel et al., 2020; Kug et al., 2020)."

2.8  Moreover, the models' agreement is weak in reproducing ozone in the lower troposphere
and the standard deviation is high in the tropics. In this context, is the conclusion that ENSO
affects the lower troposphere in the tropics convincing?

The conclusion of ENSO effects on lower tropospheric ozone is convincing. We added the following sentences to the Section 4 to discuss this point:

"Despite the limited consensus among models in replicating ozone levels in the lower troposphere, and a high standard deviation particularly in tropical regions, (Figures 1 and S1), we observed noteworthy effects of ENSO on lower tropospheric ozone (Figure 2). These results exhibit a degree of independence and are not contradictory. This is because the models' mean of annual ozone is calculated over the entire 1850-2014 period, whereas the assessment of the relationship between the ENSO and annual ozone is conducted on a year-to-year basis. Furthermore, variations in ozone are also influenced by factors beyond ENSO, including other major climate modes, cyclones, and local emissions of ozone precursors such as nitrogen oxides ($NO_x$), volatile organic compounds, and carbon monoxide (CO). Biases in simulating these factors contribute to the inconsistencies of ozone in the models, although there is consensus in simulating the connection between ENSO and ozone."

2.9    Line 116 "The significant impacts of ENSO on ozone … might be associated with the transport of ozone from east Asia". If so, the effect of ENSO on ozone over east Asia should be found. But it doesn't. Can you add some explanation about it?

**Response:** We thank the reviewer for raising this point. We added the following sentences to Section 4 to further clarify this point:

[revised manuscript text omitted]

---

## Author Response (AR2)

**1 Response to Editor's comments**

1.1 Dear Authors,

One issue raised by one of the reviewers concerns the use of climate models without coupled chemistry to evaluate the relationship between ENSO and tropospheric ozone concentrations, with which I agree. I cannot find a clear explanation for the use of those models, which do not generate ozone and are constrained by observations (that are likely highly uncertain from 1850).

In order to sort out this issue, I suggest that you only use the multi-model mean variables from models that have coupled chemistry (2,3,4,5,6,7,8, and 9 in Table 1) for Figures 1,2 and 4.

Your conclusions will be more robust if you include only 8 rather than all 12 models in the multi-model mean.

Please modify the multi-model mean to estimate the probabilities of no Granger causality and produce new plots for figures 1,2 and 4. You may leave all 12 models in Fig 3, to show the different response, if you consider valuable.

Best regards,

Graciela Raga, handling editor

**Response:** We thank editor Graciela Raga for raising this point. We modified figures 1,2 and 4 and the corresponding text. In the new figures, we only include 8 models in the computation as your suggestion. Similarly, we also modified figures S1 and S5 to S7 in the supplement.